# Potential Resistance to Oxaliplatin-Based Regimens in Gastric Cancer Patients with ERBB2 R678Q Mutation: Evidence from a National Genomic Database

**DOI:** 10.3390/cimb47060430

**Published:** 2025-06-06

**Authors:** Shuhei Suzuki, Manabu Seino, Hidenori Sato, Yosuke Saito, Koki Saito, Yuta Yamada, Koshi Takahashi, Ryosuke Kumanishi, Tadahisa Fukui

**Affiliations:** 1Department of Clinical Oncology, Yamagata University School of Medicine, 2-2-2 Iida-nishi, Yamagata 990-9585, Japan; 2Yamagata Hereditary Tumor Research Center, Yamagata University, 1-4-12 Kojirakawa, Yamagata 990-8560, Japan; 3Obstetrics and Gynecology, Yamagata University School of Medicine, 2-2-2 Iida-nishi, Yamagata 990-9585, Japan; 4Genomic Information, Yamagata University School of Medicine, 2-2-2 Iida-nishi, Yamagata 990-9585, Japan

**Keywords:** gastric cancer, ERBB2, HER2, R678Q mutation, genomic testing, oxaliplatin, chemotherapy resistance, precision medicine, biomarker, pharmacogenomics

## Abstract

Epidermal growth factor receptor 2 (*ERBB2/HER2*) is a critical biomarker in gastric cancer management, but the clinical implications of specific ERBB2 mutations remain poorly characterized. Methods/Results: We investigated the *ERBB2* R678Q mutation, utilizing the Center for Cancer Genomics and Advanced Therapeutics (C-CAT) database, which involved the analysis of 3116 gastric/gastroesophageal junction adenocarcinomas. *ERBB2* mutations were identified in 130 cases, with R678Q present in 40 patients. These patients exhibited significantly lower response rates to oxaliplatin-based regimens compared to *ERBB2* wild-type cases (19.0% vs. 38.0%, *p* = 0.03), while other *ERBB2* mutations demonstrated no such resistance. No significant differences in the response were observed to the ramucirumab or nivolumab regimens. Conclusions: Our findings suggest that the *ERBB2* R678Q mutation may predict a poor response to oxaliplatin-based therapy. This study provides real-world evidence supporting the potential clinical relevance of this specific *ERBB2* mutation in treatment decision making for gastric cancer.

## 1. Introduction

Gastric cancer remains one of the most common and lethal malignancies worldwide, ranking fifth in terms of incidence and fourth in terms of mortality among all types of cancer globally [1]. In Japan, despite declining incidence rates, gastric cancer continues to represent a significant public health burden. The management of advanced gastric cancer has evolved substantially over the past two decades, with the incorporation of targeted therapies and immune checkpoint inhibitors complementing conventional cytotoxic chemotherapy regimens [2].

*ERBB2* (*HER2*) represents one of the most clinically relevant biomarkers in gastric cancer, with approximately 15–20% of gastric and gastroesophageal junction (GEJ) adenocarcinomas demonstrating *ERBB2* overexpression or amplification [3]. The landmark ToGA trial established trastuzumab plus chemotherapy as the standard first-line treatment for *ERBB2*-positive advanced gastric cancer, demonstrating significant improvements in overall survival compared to chemotherapy alone [4]. This success led to increased interest in understanding the broader spectrum of *ERBB2* alterations beyond amplification and overexpression, including various mutations that might impact the treatment response and disease biology.

The molecular landscape of gastric cancer has been increasingly characterized by comprehensive genomic profiling efforts, revealing distinct molecular subtypes with potential therapeutic implications [5]. The Cancer Genome Atlas (TCGA) identified four molecular subtypes of gastric cancer, namely Epstein–Barr virus positive, a microsatellite instability status, genomically stable, and chromosomal instability, each associated with different patterns of molecular alterations and clinical behaviors [6,7]. The Asian Cancer Research Group (ACRG) proposed an alternative classification system with prognostic significance, further highlighting the molecular heterogeneity of this disease [8].

Within this complex molecular landscape, specific *ERBB2* mutations have emerged as potentially actionable alterations. While *ERBB2* amplification and overexpression have been extensively studied, the clinical significance of specific *ERBB2* point mutations remains less well-characterized. The *ERBB2* R678Q mutation, occurring in the juxtamembrane domain of the receptor, represents one such alteration, with emerging evidence suggesting distinct biological and clinical implications [9].

Recent preclinical studies have demonstrated that the *ERBB2* R678Q mutation may confer unique conformational changes to the receptor, potentially altering downstream signaling pathways and therapeutic vulnerabilities [9]. However, the real-world prevalence and clinical impact of this specific mutation in gastric cancer patients remain poorly understood, particularly in Asian populations, where gastric cancer incidence is highest.

The standard first-line treatment for *ERBB2*-negative advanced gastric cancer typically involves platinum-based combinations, with oxaliplatin plus fluoropyrimidines being widely used in Japanese clinical practice [10]. The SPIRITS trial and subsequent studies established S-1 plus cisplatin as an effective regimen [11], while the G-SOX trial demonstrated the non-inferiority of S-1 plus oxaliplatin compared to S-1 plus cisplatin, with improved tolerability [12]. Understanding how specific molecular alterations, such as the *ERBB2* R678Q mutation, might influence the response to these standard therapies represents a critical unmet need in regard to optimizing treatment selection.

The Center for Cancer Genomics and Advanced Therapeutics (C-CAT) in Japan has established a comprehensive national database of cancer genomic profiles, providing a unique opportunity to investigate rare molecular alterations and their clinical implications in real-world settings [13,14]. This resource allows for the exploration of relationships between specific genomic alterations and treatment outcomes across various cancer types, including gastric cancer.

In this study, we aimed to characterize the prevalence, molecular context, and treatment implications of the *ERBB2* R678Q mutation in gastric cancer patients, using the C-CAT database. We hypothesized that this specific mutation might confer distinct treatment response patterns compared to *ERBB2* wild-type tumors or those harboring other *ERBB2* mutations. Our findings provide novel insights into the clinical significance of this molecular alteration and its potential implications for treatment decision making in gastric cancer.

## 2. Materials and Methods

### 2.1. Study Cohort and Data Acquisition

The C-CAT database (ver. 20250218) constitutes Japan’s most comprehensive cancer genomics repository, encompassing extensive molecular characterization data [13,14] from 92,802 patients, spanning various cancer types. Within this database, there is a total of 5614 esophagogastric cancer patients, of which 3116 are categorized as gastric or gastroesophageal junction (GEJ) adenocarcinomas, offering a substantial dataset for investigation. The database integrates outcomes from five different genomic profiling platforms, namely the NCC Oncopanel System, FoundationOne CDx, FoundationOne Liquid CDx, Guardant360 CDx, and GenMine™ TOP Cancer Panel, each utilized during particular time intervals, spanning from June 2019 through February 2025. These platforms provide different capabilities regarding the gene scope, sample prerequisites, and supplementary molecular characteristics evaluation. The NCC Oncopanel System analyzes 114 genes and incorporates a microsatellite instability assessment in its updated versions, whereas FoundationOne CDx delivers extensive coverage of 324 genes, with a combined microsatellite instability (MSI) and tumor mutational burden (TMB) evaluation. FoundationOne Liquid CDx provides plasma-based testing, with equivalent gene coverage to its tissue-based equivalent, which is especially beneficial for patients that are incapable of undergoing a tissue biopsy. Guardant360 CDx, although more targeted with just 74 genes, concentrates on swift plasma-based analysis, whereas the GenMine™ TOP Cancer Panel delivers the most comprehensive coverage, with 723 genes (Table 1).

### 2.2. Clinical Information Gathering and Evaluation of the Outcomes

We implemented a retrospective cohort investigation that integrated comprehensive clinical, pathological, and genomic datasets. The patients’ demographic characteristics, histopathological parameters, including neoplastic cellular density and morphological subtypes, and detailed clinical specimen information were methodically documented. Concurrent data collection encompassed therapeutic interventions, response outcomes, environmental and behavioral factors, and familial cancer incidence.

An evaluation of the genomic results was conducted through the use of standardized protocols by interdisciplinary Expert Panels, comprising medical oncologists, pathologists, clinical geneticists, bioinformaticians, and primary care physicians, in accordance with the requirements specified by the Japanese healthcare reimbursement framework.

The genomic evaluation encompassed gene alterations, the tumor mutational burden (TMB), MSI, and other notable molecular markers. The pathogenicity of the variants was determined based on curated databases, such as ClinVar, OncoKB, and jMorp. Variants were deemed pathogenic if they were categorized as ‘Likely Oncogenic’ or higher in OncoKB, or met level F or above, according to the C-CAT classification, as previously described [15,16].

Treatment outcomes were recorded by site-specific physicians referencing RECIST (Response Evaluation Criteria in Solid Tumors), and classified into complete response, partial response, stable disease, progressive disease, or not assessed categories.

### 2.3. Statistical Analysis

Exploratory statistical evaluations were conducted using Microsoft Excel 2021 and Statcel 5 (OMS Publishing Inc., Saitama, Japan). The analysis aimed to examine the relationships between molecular characteristics and clinical outcomes. The categorical data were assessed using the chi-square test. The response rate was defined as the sum of complete and partial responses. Univariate analysis was performed for the available clinical variables. All the statistical procedures were performed using two-sided tests, and the findings were interpreted as exploratory, due to the small sample size characteristic of this rare tumor population. Missing values were excluded from the analysis without the use of imputation methods, consistent with the retrospective design of the study.

## 3. Results

### 3.1. Patient Characteristics

Among the 92,802 cancer cases registered in the C-CAT database (Appendix A), esophagogastric cancers represented 5614 cases (6.0%; Appendix A). Within this cohort, 3116 cases were specifically classified as gastric or gastroesophageal junction adenocarcinomas (Table 2). The number of *ERBB2* amplification cases was 503 (Appendix A), and *ERBB2* mutations were identified in 130 patients (4.1%; Table 3), with the R678Q mutation detected in 40 patients (1.3% of all gastric or gastroesophageal junction adenocarcinoma cases and 30.8% of *ERBB2* mutation cases; Table 4).

The demographic and clinical characteristics of patients with *ERBB2* R678Q mutations are summarized in Table 4. The median age was 64 years (range: 32–84 years), with most patients in the 60–69 years age group. The gender distribution indicated an approximate male-to-female ratio of 1:2, made up of 14 males and 26 females.

Regarding specimen characteristics, primary site samples (from the stomach) were most common (n = 26, 65.0%), followed by metastatic site samples (n = 7, 17.5%), and blood samples (n = 7, 17.5%). The most used predominant tests were FoundationOne CDx (n = 33, 82.5%) and FoundationOne Liquid CDx (n = 6, 15.0%).

The distribution of metastatic sites showed patterns typical of advanced gastric cancer, with peritoneal dissemination being the most frequent (52.5%), followed by lymph node metastases (27.5%) and liver metastases (12.5%). Among the relevant lifestyle factors, 59.0% of patients had past and/or current smoking history, while 13.5% reported heavy alcohol use. A family history of any cancer was registered in 86.1% of cases.

### 3.2. Overview of Genomic Testing Results

We analyzed the genetic alterations identified through cancer genome profiling tests in the 40 gastric cancer cases with ERBB2 R678Q mutations (Figure 1). The most frequently observed alterations were in the ARID1A gene, detected in 13 cases (32.5%), indicating its fundamental role in gastric carcinogenesis [17]. This was followed by TP53 alterations in 12 cases (30.0%), KRAS alterations in 9 cases (22.5%), and TGFBR2 alterations in 7 cases (17.5%).

Several other cancer-related alterations were detected at lower frequencies, including alterations in PTEN, PIK3CA, APC, and CTNNB1 genes. These alterations may influence the selection of targeted therapies and have prognostic implications. Regarding the microsatellite stability status, four cases (10.0%) were microsatellite instability high (MSI-H).

### 3.3. Assessment of Treatment Efficacy

The analysis of the treatment outcomes revealed striking differences between patients with *ERBB2* R678Q mutations and other subgroups, particularly in response to oxaliplatin-containing regimens. The objective response rate to oxaliplatin-based therapy in patients with *ERBB2* R678Q mutations was significantly lower at 19.0% (95% Confidence Interval: 7.2–30.9%) compared to 38.0% (36.0–40.3%) in *ERBB2* wild-type patients (*p* = 0.03).

Notably, patients with other *ERBB2* mutations showed no evidence of oxaliplatin resistance, with a response rate of 40.6% (29.4–52.9%) that was nearly identical to the wild-type population (*p* = 0.71). This finding suggests that the R678Q mutation specifically, rather than *ERBB2* mutations broadly, confers resistance to oxaliplatin-based therapy.

In contrast to the findings for oxaliplatin, no significant differences in the response rates were observed with ramucirumab-containing regimens between *ERBB2* R678Q mutation cases (9.1%; 95% Confidence Interval: 2.5–26.4%), other *ERBB2* mutations (13.7%; 6.8–25.8%), and wild-type cases (22.8%; 21.0–24.7%). Similarly, the response rates to nivolumab-containing therapies were comparable across *ERBB2* R678Q mutation cases (20.0%; 8.9–39.3%), other *ERBB2* mutations (18.9%; 10.5–31.9%), and wild-type cases (25.3%; 23.5–27.2%). The univariate analysis revealed no significant associations between the available clinical variables (age, sex, smoking history, alcohol consumption, metastases sites) and the oxaliplatin response in patients with *ERBB2* R678Q mutations, although the small sample size limits the interpretation of these findings.

These findings suggest that the *ERBB2* R678Q mutation confers specific resistance to oxaliplatin-based therapy, without significantly altering the sensitivity to anti-angiogenic or immune checkpoint inhibitor therapies.

## 4. Discussion

Our analysis of the C-CAT database provides important insights into the clinical significance of the *ERBB2* R678Q mutation in regard to gastric cancer treatment. The identification of this mutation in approximately 1% of gastric cancer patients represents an important finding, as these patients showed lower response rates to oxaliplatin-based therapy, which represents a cornerstone of standard treatment [12] approaches to advanced gastric cancer. While the molecular complexity of gastric cancer is becoming increasingly recognized in research, our understanding of how specific genetic alterations affect treatment outcomes is limited. Our study expands this knowledge by showing how certain *ERBB2* mutations may predict resistance to common chemotherapy regimens in clinical settings.

The disparity in the response rates between patients with *ERBB2* R678Q mutations and those with wild-type *ERBB2* or other *ERBB2* mutations raises important questions about the underlying biological mechanism(s) involved. The R678Q mutation occurs in the juxtamembrane domain of the ERBB2 receptor, a region critical for receptor dimerization and signal transduction [18]. Previous studies have suggested that mutations around the domain may alter receptor conformation, leading to constitutive activation or modified interactions with downstream signaling molecules [9]. This altered signaling might potentially upregulate DNA repair mechanisms or detoxification pathways that specifically counteract the cytotoxic effects of oxaliplatin.

Oxaliplatin exerts its cytotoxic effects primarily through the formation of DNA cross-links, leading to replication fork stalling and, ultimately, apoptosis [19,20,21]. Several mechanisms of oxaliplatin resistance have been identified, including enhanced nucleotide excision repair, increased expression of copper transporters, and the upregulation of glutathione-related detoxification pathways [22]. It is plausible that *ERBB2* R678Q mutation-induced signaling might enhance one or more of these resistance mechanisms, leading to the observed clinical phenotype.

Interestingly, we observed that this resistance was specifically limited to oxaliplatin therapy, with no measurable effect on outcomes with other treatment modalities, such as ramucirumab or nivolumab. This pattern suggests the targeted interference with specific drug mechanisms rather than an overall more aggressive cancer behavior, consistent with the growing understanding that molecular signatures can create highly selective therapeutic vulnerabilities.

Our findings may have implications for clinical practice and trial design. Currently, oxaliplatin-based combinations represent standard first-line treatment options for many patients with advanced gastric cancer, especially in East Asian countries, where S-1 plus oxaliplatin (SOX) is widely used [12]. The identification of a molecular subgroup with primary resistance to this approach suggests that alternative strategies should be considered for these patients.

The timing of comprehensive genomic profiling appears to be an important consideration, based on our analysis. Current healthcare policies, in many circumstances, including in Japan, often restrict comprehensive genomic profiling (CGP) coverage to later-line settings [23]. This approach may be counterproductive for patients with *ERBB2* R678Q mutations, as our data suggest that earlier comprehensive testing might help avoid potentially ineffective oxaliplatin-based therapy and guide more appropriate therapeutic choices from the outset. The cost effectiveness of implementing CGP before first-line therapy [24] warrants a careful evaluation, considering both the direct costs of testing and the potential savings from avoiding less optimal treatments.

Several therapeutic alternatives might be considered for patients with *ERBB2* R678Q mutations. Given the location of the mutation within the ERBB2 receptor, targeted approaches with novel *ERBB2*-directed therapies might represent promising strategies. Recent developments in regard to antibody–drug conjugates targeting *ERBB2*, such as trastuzumab deruxtecan (T-DXd), have demonstrated promising efficacy in regard to *ERBB2*-positive gastric cancer [25], and their activity in the context of specific *ERBB2* mutations warrants further investigation. Additionally, small-molecule tyrosine kinase inhibitors, with activity against mutant *ERBB2*, might offer alternative approaches [26].

The co-occurrence of *ERBB2* R678Q mutations with alterations in other genes provides an important molecular context for these tumors. The frequently observed co-alterations included *ARID1A*, *TP53*, and *KRAS* mutations. The high frequency of *TP53* alterations aligns with the chromosomal instability molecular subtype of gastric cancer previously identified in comprehensive genomic analyses [27]. Notably, *KRAS* mutations were also observed in these tumors, which may contribute to resistance mechanisms against both targeted and conventional therapies. While our current sample size precludes definitive analysis of specific co-alteration patterns and their relationship to oxaliplatin resistance, these genomic landscapes suggest that future studies should investigate whether particular combinations of alterations might further stratify the treatment response within this molecular subgroup.

Our findings also have implications for clinical trial design and biomarker development. The observed impact of *ERBB2* R678Q mutations on treatment outcomes indicates that future trials of platinum-based therapies should consider comprehensive molecular profiling in regard to their eligibility criteria or stratification factors. Additionally, the development of companion diagnostic tests that can detect specific *ERBB2* mutations may be necessary to optimize patient selection for various therapeutic approaches.

Several limitations of our study warrant discussion. First, our analysis was based on a retrospective database review without centralized pathological confirmation of the gastric origin, potentially including some distal esophageal adenocarcinomas misclassified as GEJ tumors. Second, the relatively small sample size of *ERBB2* R678Q mutation cases limits the statistical power of the findings, requiring cautious interpretation of the subgroup analyses. Third, the database relies on accurate reporting from multiple institutions, introducing potential variability in response assessments and data quality. Fourth, our study design is retrospective, with inherent selection biases related to which patients underwent comprehensive genomic profiling. Finally, we focused specifically on oxaliplatin, ramucirumab, and nivolumab responses, but did not analyze the impact of other agents, such as taxanes or fluoropyrimidines, which might provide additional insights into the specificity of the resistance phenotype. Since taxanes and fluoropyrimidines are often included in postoperative chemotherapy regimens, this creates challenges in regard to result interpretation and analysis.

Despite these limitations, our findings might provide potential evidence for the clinical relevance of *ERBB2* R678Q mutations in gastric cancer. The lower response rates observed in patients with these mutations suggest that the current standard testing methods could be expanded to include the detection of specific *ERBB2* mutations for optimal patient selection for oxaliplatin-based therapy. The implementation of comprehensive genomic profiling before first-line therapy might contribute to improved treatment outcomes through more precise treatment allocation.

Future research directions suggested by our findings include mechanistic studies to elucidate the biological basis of oxaliplatin resistance in *ERBB2* R678Q-mutated tumors, prospective validation of alternative treatment approaches for this molecular subgroup, and an evaluation of specific *ERBB2*-targeted therapies in this context. Additionally, the development of cost-effective, targeted testing approaches for key actionable mutations, including *ERBB2* R678Q, could help bridge the gap between standard testing and comprehensive genomic profiling in resource-constrained settings.

## 5. Conclusions

This analysis of the C-CAT database suggests that the *ERBB2* R678Q mutation, present in approximately 1% of gastric cancer cases, is associated with a significantly reduced response to oxaliplatin-based therapy. This resistance appears specific to this particular mutation rather than ERBB2 mutations more broadly, and does not extend to other therapeutic approaches, such as ramucirumab or nivolumab. The current practice of limiting comprehensive genomic profiling to later-line settings may result in suboptimal treatment selection for patients harboring this mutation. Our findings support the implementation of more comprehensive molecular testing before first-line therapy to improve patient selection and treatment outcomes. Further research is needed to elucidate the biological mechanisms underlying this resistance phenotype and to develop effective alternative therapeutic strategies for this molecular subgroup of gastric cancer patients.

## Figures and Tables

**Figure 1 cimb-47-00430-f001:**
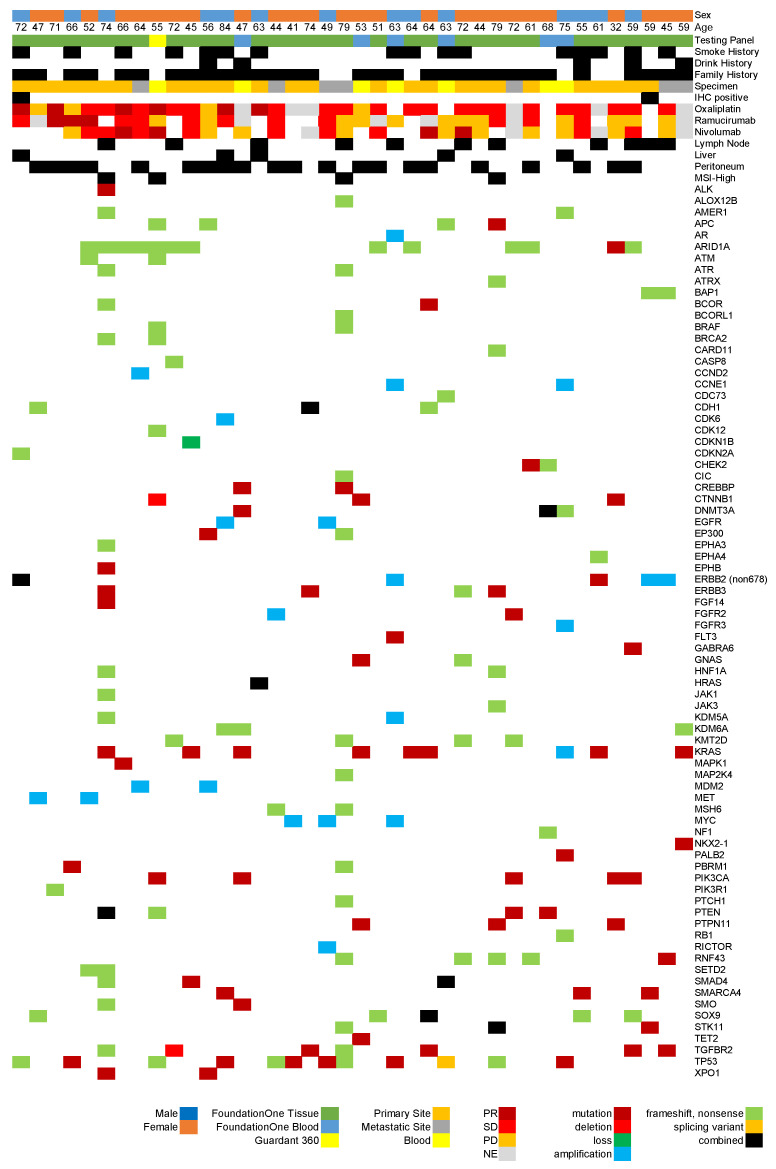
Genomic findings and chemotherapeutic treatment responses in patients harboring *ERBB2* R678Q mutations registered in the Center for Cancer Genomics and Advanced Therapeutics database. Treatment responses were evaluated according to the RECIST criteria and classified as PR (partial response), SD (stable disease), PD (progressive disease), or NE (not evaluated). The figure illustrates the genomic landscape and therapeutic outcomes in gastric cancer patients harboring *ERBB2* R678Q mutations, demonstrating the co-alteration patterns that may inform future treatment strategies.

**Table 1 cimb-47-00430-t001:** Catalog of genomic assays reimbursed through public health insurance in Japan (solid malignancies).

Characteristic	FoundationOne CDx	FoundationOne Liquid CDx	NCC Oncopanel System	Guardant 360 CDx	GenMine TOP Cancer Panel
Specimen Type	FFPE tissue	Blood	FFPE tissue	Blood	FFPE tissue
Number of Genes	324	324	114	74	723
MSI Assessment	Yes	Yes	Yes *	Yes	No
Paired Analysis	No	No	Yes	No	Yes
TMB Evaluation	Yes	Yes	Yes	No	Yes
Minimal Neoplastic Content Required	20%	N/A	20%	N/A	20%
Necessary DNA Input	50 ng	2 tubes	50 ng	2 tubes **	50 ng

* Previous versions did not incorporate MSI assessment capability. ** Including a backup tube. FFPE: Formalin-Fixed Paraffin-Embedded, MSI: microsatellite instability, TMB: tumor mutational burden, N/A: not available.

**Table 2 cimb-47-00430-t002:** Gastric or gastroesophageal junction adenocarcinomas, background of the patients registered in the Center for Cancer Genomics and Advanced Therapeutics database.

Gastric or Gastroesophageal Junction Adenocarcinoma Cases (n = 3116)			
Cancer Genomics Test		Treatment Response to Oxaliplatin *	
FoundationOne CDx	2262	Complete Response	28
FoundationOne Liquid CDx	394	Partial Response	813
NCC Oncopanel System	305	Stable Disease	815
GenMine^TM^ TOP Cancer Panel	96	Progressive Disease	549
Guardant360 CDx	59	Not Evaluated	379
Sex		Treatment Response to Ramucirumab *	
Male	5104	Complete Response	10
Female	1012	Partial Response	387
Age Group (years)		Stable Disease	828
70–79		Progressive Disease	553
60–69	992	Not Evaluated	545
50–59	908	Treatment Response to Nivolumab *	
40–49	549	Complete Response	18
30–29	339	Partial Response	451
80–89	146	Stable Disease	650
20–29	129	Progressive Disease	733
10–19	46	Not Evaluated	383
90–	6		

*: Regimen containing indicated drug. The study periods for each instance of genomic testing were as follows: NCC Oncopanel System (1 June 2019 to 14 February 2025), FoundationOne^®^ CDx (1 June 2019 to 15 February 2025), FoundationOne^®^ Liquid CDx (1 August 2021 to 15 February 2025), Guardant360^®^ CDx (24 July 2023 to 14 February 2025), and GenMineTM TOP Cancer Panel (1 August 2023 to 14 February 2025).

**Table 3 cimb-47-00430-t003:** Gastric or gastroesophageal junction adenocarcinomas with *ERBB2* mutation, background of registered cases in the Center for Cancer Genomics and Advanced Therapeutics database.

Gastric or Gastroesophageal Junction Adenocarcinoma *ERBB2* Mutation Cases (N = 130)			
Cancer Genomics Test		Treatment Response to Oxaliplatin *	
FoundationOne CDx	103	Complete Response	2
FoundationOne Liquid CDx	19	Partial Response	29
NCC Oncopanel System	3	Stable Disease	33
GenMine^TM^ TOP Cancer Panel	3	Progressive Disease	27
Guardant360 CDx	2	Not Evaluated	16
Sex		Treatment Response to Ramucirumab *	
Male	60	Complete Response	0
Female	70	Partial Response	9
		Stable Disease	32
Age Group (years)		Progressive Disease	553
70–79	42	Not Evaluated	19
60–69	35	Treatment Response to Nivolumab *	
50–59	25	Complete Response	1
40–49	21	Partial Response	15
80–89	5	Stable Disease	23
30–39	2	Progressive Disease	40
		Not Evaluated	16

*: Regimen containing indicated drug. The study periods for each instance of genomic testing were as follows: NCC Oncopanel System (1 June 2019 to 14 February 2025), FoundationOne^®^ CDx (1 June 2019 to 15 February 2025), FoundationOne^®^ Liquid CDx (1 August 2021 to 15 February 2025), Guardant360^®^ CDx (24 July 2023 to 14 February 2025), and GenMineTM TOP Cancer Panel (1 August 2023 to 14 February 2025).

**Table 4 cimb-47-00430-t004:** Gastric or gastroesophageal junction adenocarcinomas with *ERBB2* R678Q, background of the patients registered in the Center for Cancer Genomics and Advanced Therapeutics database.

Gastric or Gastroesophageal Junction Adenocarcinoma *ERBB2* E678Q Cases (N = 40)			
**Age Group (Years; Median 62)**		**Cancer Testing Panel**	
60–69	11	FoundationOne CDx	33
70–79	10	FoundationOne Liquid CDx	6
50–59	9	Guardant360 CDx	1
40–49	8		
30–39	1	**Treatment Response to Oxaliplatin ***	
80–89	1	Complete Response	0
		Partial Response	5
**Sex**		Stable Disease	17
Female	26	Progressive Disease	5
Male	14	Not Evaluated	8
**Smoking History**		**Treatment Response to Ramucirumab ***	
No	23	Complete Response	0
Yes	16	Partial Response	2
Unknown	1	Stable Disease	10
		Progressive Disease	10
**Drinking History**		Not Evaluated	7
No	32		
Yes	5	**Treatment Response to Nivolumab ***	
Unknown	3	Complete Response	0
		Partial Response	5
**Metastatic Sites**		Stable Disease	10
Peritoneum	21	Progressive Disease	10
Lymph Node	11	Not Evaluated	3
Liver	5		

*: Regimen containing indicated drug. The study periods for each instance of genomic testing were as follows: NCC Oncopanel System (1 June 2019 to 14 February 2025), FoundationOne^®^ CDx (1 June 2019 to 15 February 2025), FoundationOne^®^ Liquid CDx (1 August 2021 to 15 February 2025), Guardant360^®^ CDx (24 July 2023 to 14 February 2025), and GenMineTM TOP Cancer Panel (1 August 2023 to 14 February 2025).

## Data Availability

The dataset generated during this study are not publicly accessible due to confidentiality agreements as part of the ethics approval process.

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
