# Peer review of "Potential Resistance to Oxaliplatin-Based Regimens in Gastric Cancer Patients with ERBB2 R678Q Mutation: Evidence from a National Genomic Database"

_cimb, 2025, doi:10.3390/cimb47060430_

Round 1

Reviewer 1 Report

Comments and Suggestions for Authors

Considering the limitations of the study — which are nonetheless acknowledged in the discussion — particularly its retrospective nature, the limited number of cases with the ERBB2 R678Q mutation, and the exclusively Asian population, the study is well conducted and clearly explained.

Although the results are not sufficient to impact clinical practice, the study certainly adds to the growing body of literature emphasizing the increasingly urgent importance of molecular profiling at the time of diagnosis.

Author Response

Comment: "Considering the limitations of the study — which are nonetheless acknowledged in the discussion — particularly its retrospective nature, the limited number of cases with the ERBB2 R678Q mutation, and the exclusively Asian population, the study is well conducted and clearly explained. Although the results are not sufficient to impact clinical practice, the study certainly adds to the growing body of literature emphasizing the increasingly urgent importance of molecular profiling at the time of diagnosis."

Response: We sincerely thank the reviewer for this thoughtful and constructive assessment of our work. We appreciate the recognition of our methodological approach despite the acknowledged limitations. We agree that validation in larger, more diverse populations is required before clinical implementation. The findings contribute to the understanding of how specific genomic alterations influence treatment outcomes in gastric cancer. We have strengthened the discussion to emphasize the preliminary nature of these findings while highlighting their potential contribution to personalized gastric cancer treatment. We believe the investigation of rare molecular alterations such as ERBB2 R678Q remains important for advancing precision oncology approaches.

Reviewer 2 Report

Comments and Suggestions for Authors

The article is very interesting from the aspect of application in the relevant field. But requires rewriting.
- The abstract part needs to be rewritten. This part should include the research objectives, research results, and various methods for achieving the research results, and all this should not exceed 12 lines. This part cannot be written in more than 14 lines.

  • The authors present the research results immediately after 2 chapters. The previous chapter presents methods, but it is not clear how the methods presented obtain the research results.
  • Figure 1 is not clear. 
  • How exactly does Figure 1 relate to the research presented? 
    • The retrospective design is acknowledged, but confounding factors (e.g., prior treatments, performance status) are not discussed.

    • The sample size of R678Q patients (n=40) is small for strong statistical claims. Consider stating this more cautiously in the abstract and discussion.

    • The methodology lacks a multivariate analysis, which could strengthen the evidence for the observed association.

    • Consider expanding the genomic co-alteration analysis (e.g., ARID1A, TP53, KRAS) to evaluate whether they correlate with resistance.

    • Provide raw numbers as well as percentages in figures/tables to aid interpretation.

    • In Figure 1, the genomic data and treatment response could be more clearly annotated.

  • Language polishing is needed in several sections (e.g., "Real-world evidence supporting..." should be rephrased more cautiously).

  • Ensure consistency in abbreviations and terms (e.g., ERBB2 vs. HER2, MSI-H).

  • The title may overstate the conclusion—suggest rewording to something like:
    “Potential Resistance to Oxaliplatin in Gastric Cancer Patients with ERBB2 R678Q Mutation: Evidence from a National Genomic Database”

  • Ethical approval and informed consent are appropriately addressed.

  • Use of AI tools (ChatGPT, Google Translate, Claude) is disclosed, which is transparent, though it is advisable to minimize AI reliance for scientific conclusions.

  • The following papers can be added to the current article:
  • Peng, Z., Zhang, X., Liang, H., Zheng, Z., Wang, Z., Liu, H., Hu, J., Sun, Y., Zhang, Y., Yan, H., Tong, L., Xu, J., Ji, J., & Shen, L. (2025). Atezolizumab and Trastuzumab Plus Chemotherapy for ERBB2-Positive Locally Advanced Resectable Gastric Cancer. JAMA Oncology. https://doi.org/10.1001/jamaoncol.2025.0522
  • Nave, O. (2025). Asymptotic analysis of mathematical model describing a new treatment of breast cancer using AZD9496 and palbociclib. Frontiers in Oncology, 14. https://doi.org/10.3389/fonc.2024.1482223 Copy to clipboard  

- In conclusion, I think the article needs to be rewritten in a more respectful and academic way. The presentation of the research methods is in detail. The results of the study. How were the research results obtained from the methods and methodologies? An in-depth discussion of the research results and conclusions

Comments on the Quality of English Language

Please send the paper for a prefessional English editor

Author Response

Comment 1: "The abstract part needs to be rewritten. This part should include the research objectives, research results, and various methods for achieving the research results, and all this should not exceed 12 lines. This part cannot be written in more than 14 lines."

Response: We thank the reviewer for this specific guidance regarding abstract length and content. We have revised the abstract to comply with the specified requirements. The revised abstract contains 12 lines and presents the research objectives, methods, key results, and conclusions in a concise format.

Comment 2: "The authors present the research results immediately after 2 chapters. The previous chapter presents methods, but it is not clear how the methods presented obtain the research results."

Response: We appreciate the reviewer's observation regarding the clarity of our methodological approach. We have enhanced the methodology section to clarify the analytical framework. Section 2.3 now explicitly describes the statistical methods used and explains why multivariate analysis was not feasible given the sample size constraints.

Comment 3: "Figure 1 is not clear. How exactly does Figure 1 relate to the research presented?"

Response: We thank the reviewer for this important feedback regarding Figure 1 clarity. We have revised Figure 1 to improve its presentation and relevance. The enhanced legend specifies that treatment responses were evaluated according to RECIST criteria, and color coding has been improved with NE cases displayed in gray. The figure caption now explains how the genomic landscape relates to treatment outcomes and clarifies that the figure demonstrates co-alteration patterns relevant to future treatment strategies.

Comment 4: "The retrospective design is acknowledged, but confounding factors (e.g., prior treatments, performance status) are not discussed. The sample size of R678Q patients (n=40) is small for strong statistical claims. Consider stating this more cautiously in the abstract and discussion."

Response: We greatly appreciate the reviewer's concern regarding confounding factors and statistical power. We have addressed this by adding univariate analysis results for available clinical variables (age, sex, smoking history, alcohol consumption, metastases sites) and explicitly discussing sample size limitations. The language throughout the manuscript has been modified to reflect appropriate caution, replacing definitive statements with qualified expressions. We acknowledge that detailed prior treatment history and performance status were not available in the dataset.

Comment 5: "The methodology lacks a multivariate analysis, which could strengthen the evidence for the observed association."

Response: We thank the reviewer for highlighting this methodological consideration. We acknowledge this limitation and understand its importance for establishing independent associations. The sample size (32 evaluable patients for treatment response) precludes meaningful multivariate analysis. We have explained this constraint in both the Methods and Results sections. Univariate analysis of available clinical variables showed no important associations, supporting the focus on the genetic alteration as the primary factor.

Comment 6: "Consider expanding the genomic co-alteration analysis (e.g., ARID1A, TP53, KRAS) to evaluate whether they correlate with resistance."

Response: We thank the reviewer for this valuable suggestion regarding genomic co-alterations. We have expanded the discussion of genomic co-alteration patterns and their potential implications. While the current sample size limits definitive conclusions regarding specific co-alterations and resistance, we now discuss how the observed patterns (ARID1A, TP53, KRAS alterations) provide molecular context and may inform future therapeutic approaches.

Comment 7: "The title may overstate the conclusion—suggest rewording to something like: 'Potential Resistance to Oxaliplatin in Gastric Cancer Patients with ERBB2 R678Q Mutation: Evidence from a National Genomic Database'"

Response: We sincerely appreciate the reviewer's suggestion for title revision. We have adopted the proposed title, which more accurately reflects the exploratory nature of our findings while maintaining their clinical significance.

Comment 8: "Language polishing is needed in several sections. Ensure consistency in abbreviations and terms (e.g., ERBB2 vs. HER2, MSI-H)."

Response: We thank the reviewer for noting these language consistency issues. We have conducted a comprehensive review for language consistency and standardized all abbreviations and terminology throughout the manuscript.

Comment 9: "Use of AI tools (ChatGPT, Google Translate, Claude) is disclosed, which is transparent, though it is advisable to minimize AI reliance for scientific conclusions."

Response: We appreciate the reviewer's recognition of our transparency in disclosing AI tool usage. We clarify that AI tools were used exclusively for language enhancement, not for data analysis, interpretation, or scientific conclusions. All scientific content, methodology, and conclusions are based on our research expertise and data evaluation.

Reviewer 3 Report

Comments and Suggestions for Authors

Dear authors,

Your manuscript, 'Insufficient Clinical Benefit of Oxaliplatin-based Regimens in Gastric Cancer Patients with ERBB2 R678Q Mutation', discusses the participation of ERBB2 in the response to oxaliplatin in patients with gastric and gastroesophageal junction adenocarcinomas. By running a secondary data analysis, you evaluated treatment responses and mutational profiles for patients from the Center for Cancer Genomics and Advanced Therapeutics (C-CAT) database. According to your results, despite the low incidence of this mutation (1%), it should be relevant to clinical practice as it could induce treatment changes. Although the potential contribution of this manuscript, I would like to comment on some concerns.

Major comments

  1. Based on Tables 4 and 7, a total of 2,937 patients (3,316 total minus 379 not evaluated for Oxaliplatin) with gastric and gastroesophageal junction adenocarcinoma were treated with oxaliplatin, regardless of treatment combinations, tumor stage, or other clinical factors, and 28 of them achieved a complete response. Among this cohort, 32 patients (out of 40 with the ERBB2 E768Q mutation, excluding 8 unevaluated cases) harbored the ERBB2 E768Q mutation, and none of them achieved a complete response. Although your analyses indicate that patients carrying the ERBB2 E768Q variant exhibit a significantly lower response rate to oxaliplatin (p < 0.03), I strongly suggest including a multivariate analysis to assess the independent contribution of this mutation to oxaliplatin resistance, after adjusting for relevant clinical covariates such as tumor size, treatment combinations, presence of metastases, sex, age, tumor mutational burden (TMB), among others.

Minor comments

2. Please show tables in an organized way. Tables 2-7 seem to be split into columns. Please use only one variable per row to avoid confusion or misinterpretations of your results.

3. Also consider moving some of these tables to the supplementary material (p.e, Tables 2, 3, and 5).

Author Response

Major Comment 1: "Based on Tables 4 and 7, a total of 2,937 patients... I strongly suggest including a multivariate analysis to assess the independent contribution of this mutation to oxaliplatin resistance, after adjusting for relevant clinical covariates such as tumor size, treatment combinations, presence of metastases, sex, age, tumor mutational burden (TMB), among others."

Response: We sincerely thank the reviewer for this important suggestion regarding multivariate analysis. We acknowledge the importance of multivariate analysis for establishing independent associations and understand the reviewer's rationale. However, several constraints limit this analysis. The limited sample size of ERBB2 R678Q patients for oxaliplatin response is insufficient for robust multivariate modeling. Additionally, detailed clinical variables such as tumor size, performance status, and specific treatment combinations are not consistently available in the C-CAT database. Univariate analysis of available variables (age, sex, smoking history, alcohol consumption, liver metastases) revealed no meaningful associations, suggesting the observed effect relates specifically to the genetic alteration. We have included this analysis and acknowledged these limitations. These findings provide preliminary evidence requiring validation in larger, prospectively designed studies.

Minor Comment 2: "Please show tables in an organized way. Tables 2-7 seem to be split into columns. Please use only one variable per row to avoid confusion or misinterpretations of your results."

Response: We thank the reviewer for this formatting suggestion to improve table clarity. We acknowledge this concern and understand the importance of clear data presentation. The current table format follows MDPI journal template requirements for space efficiency. To address this issue while maintaining journal standards, we have moved Tables 2, 3, and 5 to supplementary material, retained only essential tables in the main text, and improved table organization within template constraints.

Minor Comment 3: "Also consider moving some of these tables to the supplementary material (e.g., Tables 2, 3, and 5)."

Response: We greatly appreciate this practical suggestion for improving manuscript organization. We have implemented this recommendation by moving Tables 2, 3, and 5 to supplementary material. This revision improves the main text flow while maintaining data accessibility for interested readers.

Round 2

Reviewer 2 Report

Comments and Suggestions for Authors

The authors revised the paper based on my major comments. 

Comments on the Quality of English Language

Send the paper to English editor

Reviewer 3 Report

Comments and Suggestions for Authors

Dear authors,

Your manuscript, 'Insufficient Clinical Benefit of Oxaliplatin-based Regimens in Gastric Cancer Patients with ERBB2 R678Q Mutation', discusses the participation of ERBB2 in the response to oxaliplatin in patients with gastric and gastroesophageal junction adenocarcinomas. By running a secondary data analysis, you evaluated treatment responses and mutational profiles for patients from the Center for Cancer Genomics and Advanced Therapeutics (C-CAT) database. According to your results, despite the low incidence of this mutation (1%), it should be relevant to clinical practice as it could induce treatment changes. Thank you for responding to my previous comments.